# Characteristics of Antibiotic Resistance and Tolerance of Environmentally Endemic *Pseudomonas aeruginosa*

**DOI:** 10.3390/antibiotics11081120

**Published:** 2022-08-18

**Authors:** Seryoung Kim, Satomi Masai, Keiji Murakami, Momoyo Azuma, Keiko Kataoka, Mayu Sebe, Kazuya Shimizu, Tomoaki Itayama, Niwooti Whangchai, Kanda Whangchai, Ikko Ihara, Hideaki Maseda

**Affiliations:** 1Biomedical Research Institute, National Institute of Advanced Industrial Science and Technology, 1-8-31 Midorigaoka, Osaka 563-8577, Japan; 2Department of Clinical Nutrition, Faculty of Health Science and Technology, Kawasaki University of Medical Welfare, 288 Matsushima, Okayama 701-0193, Japan; 3Department of Infection Control and Prevention, Tokushima University Hospital, 2-50-1 Kuramoto, Tokushima 770-8503, Japan; 4Department of Microbiology and Genetic Analysis, Institute of Biomedical Science, Tokushima University Graduate School, Tokushima 770-8503, Japan; 5Faculty of Life Sciences, Toyo University, 1-1-1 Izumino, Itakura-machi, Ora-gun, Gunma 374-0193, Japan; 6Graduate School of Engineering, Nagasaki University, 1-14 Bunkyo, Nagasaki 852-8131, Japan; 7Faculty of Fisheries Technology and Aquatic Resources, Maejo University, Chiang Mai 50290, Thailand; 8Center of Excellence in Bioresources for Agriculture, Industry and Medicine, Chiang Mai University, Chiang Mai 50200, Thailand; 9Department of Agricultural Engineering and Socio-Economics, Kobe University, Kobe 657-8501, Japan; 10Graduate School of Life and Environmental Sciences, University of Tsukuba, 1-1-1 Tennodai, Ibaraki 305-8577, Japan; 11Department of Environmental Engineering and Green Technology, Malaysia-Japan International Institute of Technology, Universiti Teknologi Malaysia, Kuala Lumpur 54100, Malaysia

**Keywords:** *Pseudomonas aeruginosa*, antibiotic-tolerance, antibiotic-resistance, environment

## Abstract

Antibiotic-resistant bacteria remain a serious public health threat. In order to determine the percentage of antibiotic-resistant and -tolerant *Pseudomonas aeruginosa* cells present and to provide a more detailed infection risk of bacteria present in the environment, an isolation method using a combination of 41 °C culture and specific primers was established to evaluate *P. aeruginosa* in the environment. The 50 strains were randomly selected among 110 isolated from the river. The results of antibiotic susceptibility evaluation showed that only 4% of environmental strains were classified as antibiotic-resistant, while 35.7% of clinical strains isolated in the same area were antibiotic-resistant, indicating a clear difference between environmental and clinical strains. However, the percentage of antibiotic-tolerance, an indicator of potential resistance risk for strains that have not become resistant, was 78.8% for clinical strains and 90% for environmental strains, suggesting that *P. aeruginosa*, a known cause of nosocomial infections, has a high rate of antibiotic-tolerance even in environmentally derived strains. It suggested that the rate of antibiotic-tolerance is not elicited by the presence or absence of antimicrobial exposure. The combination of established isolation and risk analysis methods presented in this study should provide accurate and efficient information on the risk level of *P. aeruginosa* in various regions and samples.

## 1. Introduction

Antibiotic-resistant bacteria are a serious public health problem. In particular, it is reported that the number of deaths caused by multidrug-resistant bacteria worldwide reaches 700,000 a year, and unless special measures are taken, the number of deaths per year could increase to 10 million by 2050 [1,2]. It has been reported that the main factors that lead to the acquisition of antibiotic-resistance from the outside of bacteria cells are as follows: (i) transduction [3], (ii) conjugation [4,5], (iii) transformation [6], and (iv) horizontal gene transfer mediated by OMVs [7]. In addition, mutations that reduce the binding of antibiotics to their targets [8,9,10] and increased expression of efflux pumps [11,12,13,14] that export antibiotics out of cells have been identified as mechanisms of antibiotic-resistance due to change inside of bacteria cells. Recently, however, it has become clear that apart from this mechanism, bacteria have an “antibiotic-tolerance” function, a means of surviving antibiotic exposure [15]. Antibiotic-tolerance has been identified as the ability of non- or low-growing bacteria to tolerate antibiotics when exposed to them [16], and it has recently been shown that only some bacteria in a population exhibit this property [17,18,19]. Thus, antibiotic-tolerance is a universal ability that bacteria exhibit when they form a bacterial population, but it has not been the focus of much attention, even though the concept was proposed by Bigger in 1944 in staphylococcus [20]. The presence of such bacteria in the population has been reported to be one of the major causes of treatment failure [21,22,23], and the spread of bacteria that readily demonstrate this ability results in an even greater risk of infection. Recently, Murakami’s group analyzed *Pseudomonas aeruginosa* isolated from clinical settings and reported that about 60% of the strains were high tolerance [24].

Since antibiotic-tolerance is an intrinsic property of bacteria, it is thought that high tolerance strains exist in similar proportions regardless of location, but it is possible that exposure to antibiotics during treatment is involved in the expression of this function. In order to clarify the mechanism of tolerance, it is considered necessary to further analyze tolerance in strains that are thought to have no historical exposure to antibiotics present in the environment, but there are no examples of such analyses. Therefore, in this study, *Pseudomonas aeruginosa* cells were isolated from a river flowing through the same area analyzed by the Murakami group [24] without the use of antibiotics, and the rate and status of antibiotic-tolerance were examined in detail not only to confirm the public health risk of environmental strains but also to investigate the involvement of antimicrobial exposure history in the development of tolerance.

## 2. Materials and Methods

### 2.1. Materials for Culture Medium

Luria-Bertani (LB) broth containing 10 g tryptone (Nacalai tesque, Kyoto, Japan), 5 g yeast extract (Nacalai tesque, Kyoto, Japan), and 5 g sodium chloride (Wako, Japan) per liter (pH 7.0 to 7.2) was used. Agar powder (Nacalai tesque, Kyoto, Japan) 15 g was added to the LB broth to make LB agar. Nalidixic acid cetrimide (NAC) agar was mixed with 500 mL autoclaved solution containing 20 g bacto peptone (Becton, Dickinson and Company, Sparks, MD, USA), 15 g agar powder, and 500 mL solution (pH 7.4) containing 0.3 g potassium dihydrogen phosphate and 0.2 g magnesium sulfate. We added 0.2 g cetrimide (Wako, Osaka, Japan) and 0.015 g nalidixic acid (Sigma, MO, USA) just before use. The PAO1S strain was used as a standard strain of *P. aeruginosa* [25]. The 8380 strain as a representative of clinical isolates that clarified genomic sequence was used [26].

### 2.2. Isolation of Strains from the Environment

The river water was collected from three major riversides in downtown Tokushima, Tokushima Prefecture through June to September in 2012. A nitrocellulose filter (ADVANEC^®^, Tokyo, Japan, 0.45 µm) was used to filter 500 mL of river water, which was subsequently placed on NAC agar and cultured at 37 °C or 41 °C for 24 h. The cultured colonies were spotted on new NAC agar with sterilized toothpicks, and the new agar was cultured at an appropriate temperature for 48 h. To separate single colonies, each colony was streaked onto a new NAC agar and cultured at an appropriate temperature for 24 h.

The isolated strains were cultured in LB broth for 14 h with shaking, diluted 100 times with sterilized water, and boiled for 5 min for use as a template. We then amplified 16S rDNA by PCR using the BSF 8/20: 5′-AGAGTTTGATC CTGGCTCAG-3′, BSR 1541/20: 5′-AAGGAGGTGATCCAGCCGCA-3′, or 16S PAF 139/21: 5′-GGGATAACGTCCGGAAACGGG-3′ primers. The PCR mixture contained Takara LA Taq DNA polymerase (Takara, Shiga, Japan), and the Astec PC320 thermal cycler (Astec, Fukuoka, Japan) was set as follows: 94 °C for 3 min, 30 cycles of 94 °C for 30 s, 65 °C for 1.5 min, and finally 70 °C for 10 min. The PCR products were treated using the *Hae*III restriction enzyme (Takara, Shiga, Japan) at 37 °C for 2 h.

### 2.3. Isolation of P. aeruginosa from Clinics

Clinically isolated strains of *P. aeruginosa* were obtained from Dr. Murakami and M.D. Azuma of Tokushima University Hospital (CL01 to CL42). *P. aeruginosa* clinical strains were isolated in Tokushima University Hospital and this study was performed in accordance with the Declaration of Helsinki. The study protocol was approved by the Institutional Review Board of Tokushima University Hospital (approval number: 1300-1). Strain identification was carried out by an automatic biochemical identification system (MicroScan WalkAway, Beckman Coulter, Inc., Brea, CA, USA). *P. aeruginosa* clinical strains were isolated from April to November in 2011. The sources of them were listed in the Appendix A.

### 2.4. Antibiotic Susceptibility Test for Planktonic Bacteria

The minimum inhibitory concentrations (MIC) of several antibiotics were determined by the agar dilution method using Muller-Hinton agar II (Difco, Franklin Lake, NJ, USA) according to the method recommended by the Japanese Society of Chemotherapy. The antibiotics used were chloramphenicol 8–1024 µg/mL (Wako, Japan), ciprofloxacin 0.0625–32 µg/mL (LKT Laboratories, St Paul, MN, USA), tetracycline hydrochloride 0.5–128 µg/mL (Wako, Japan), amikacin sulfate 0.25–128 µg/mL (Wako, Osaka, Japan), aztreonam 0.25–32 µg/mL (MP Biomedicals, USA), imipenem 0.125–32 µg/mL (MSD K.K., Tokyo, Japan), rifampicin 2–512 µg/mL (Tokyo Chemical Industry, Tokyo, Japan), and minocycline hydrochloride 1–256 µg/mL (Tokyo Chemical Industry, Tokyo, Japan). The standard laboratory strain, PAO1S, and representative clinical isolated strain, 8380 strain, were used as control. All experiments were performed in triplicate.

### 2.5. Carbapenem Susceptibility Test for Adherent Bacteria

Susceptibility testing for the adherent bacteria assay following the method of Murakami et al. [24]. The bacterial suspension which concentration of 2 × 10^6^ CFU/mL was divided into each well of 96-well plates (Falcon 3047, Becton Dickinson, Lincoln Park, NJ, USA; each well contained 50 µL of bacterial suspension) and then centrifuged at 450× *g* for 15 min at 25 °C. The plate was incubated at 37 °C for 1 h, then saline was removed and 100 µL of serially diluted imipenem (Wako Pure Chemical Industries, Ltd., Osaka, Japan) solutions were transferred to each well in the plate. The final imipenem concentration was used from 0.125 µg/mL to 256 µg/mL. MIC^AD^ was determined as the lowest concentration of antibiotic at which there was no bacterial growth after 24 h incubation at 37 °C. The antibiotic solutions were removed, and 200 µL of fresh LB medium was added to each well. MBC^AD^ was determined as the lowest concentration of antibiotic at which there was no bacterial growth after 24 h incubation at 37 °C. Comparison between the populations was performed by Pearson’s chi-square test. Statistical analysis was performed by JMP software 13 (SAS Institute, Inc., Tokyo, Japan) and a *p* value of <0.01 was considered as statistically significant. 

### 2.6. Genotype Analysis by PCR-Based ORF Typing (POT)

A single colony of each strain was cultured in 5 mL LB broth at 37 °C for 15 h with shaking. Each cell suspension (10 µL) was mixed with 110 µL of Cica Geneus^®^ DNA Extraction Reagent (Kanto Chemical, Tokyo, Japan), and DNA was extracted according to the manufacturer’s instructions. Multiplex PCR was performed using the Cica Geneus^®^ Pseudo POT KIT manual (Kanto Chemical, Tokyo, Japan), and band patterns were analyzed by agarose gel electrophoresis [27].

### 2.7. Evaluation of Resistance Risk

A single colony of each strain was cultured in 5 mL LB broth at 37 °C or 14 h with shaking. The cell suspensions (100 µL) were cultured on five rifampicin-containing (300 µg/mL) LB agar plates at 37 °C for 48 h. To count the live cells, the suspension was diluted 10^6^ to 10^7^ times and then applied to three LB agar plates (100 µL each); colonies were counted after 12 h. The mutation frequency was obtained by dividing the number of colonies confirmed after 48 h of culture with rifampicin containing LB agar by the number of live bacteria. The standard laboratory strain, PAO1S, was0 used as control. All experiments were performed in triplicate.

## 3. Results

### 3.1. Construction of the Method for Isolating P. aeruginosa Cells from Environments

In order to correctly assess the percentage of resistant strains in the environment and the risk, it is necessary to isolate dozens of *P. aeruginosa* cells from the environment without the use of antibiotics. Therefore, following the method of Lilly and Lowbury [28], a simple isolation method for *P. aeruginosa*, we attempted to isolate *P. aeruginosa* cells by NAC agar plates, taking advantage of their innate cetrimide resistance ability, a characteristic of the *P. aeruginosa* cells. From these, 16 colonies were randomly selected to amplify the 16S rDNA region using the universal primers for 16S rRNA, BSF 8/20 and BSR 1541/20, resulting in 9 samples with the expected 16S DNA band length. The bands were then digested with *Hae* III, one of the most frequently used restriction enzymes in RFLP analysis [29], and the cleavage pattern of the amplified band was compared to that of PAO1S, a standard strain of *P. aeruginosa*. The results showed that the cleavage band patterns of the nine isolates were identical but were very similar to the *Hae* III cleavage band pattern of *P. fluorescens* (right side of Figure 1), unlike the 16S rDNA cleavage band pattern of *P. aeruginosa*. In fact, sequencing confirmed that all nine strains were *P. fluorescens*. Therefore, we decided to develop a new isolation method since it was difficult to isolate *P. aeruginosa* by cetrimide selection alone. Since *P. aeruginosa* can grow at higher temperatures than *P. fluorescens* [30], we attempted to isolate *P. aeruginosa* cells again on NAC agar plates with the culture temperature set at 41 °C. As a result, 48 new colonies were successfully isolated from 1 L of river water collected from the same location. The genomic DNA from all of these cells, PCR and restriction enzyme analysis revealed that all of these 41 colonies showed restriction enzyme patterns identical to the cleavage band patterns of *P. aeruginosa* cells (Appendix A). Unfortunately, however, the 16S rDNA sequencing [31] revealed that although 29 isolates were 100% identical to *P. aeruginosa*, the remaining 12 isolates were found to be *P. otitidis*, suggesting that further improvement is needed.

Next, in order to efficiently distinguish *P. aeruginosa* from *P. otitidis*, we designed a new primer, 16S PAF 139/21, for the region from 139 nt to 159 nt of the 16S rDNA sequence with the largest sequence difference (Figure 2a). PAF 139/21 and the universal primer BSR 1541/20 were used to examine whether or not *P. aeruginosa* and *P. otitidis* could be distinguished by amplification. As shown in Figure 2b, DNA amplification was observed only when the genomic DNA of *P. aeruginosa* cells was used as a template. Thus, the combination of 41 °C incubation and PCR with specific primers, in addition to cetrimide, made it possible to efficiently isolate *P. aeruginosa* cells in the environment.

### 3.2. Isolation of P. aeruginosa from River Water

With the isolation method established, we set out to further isolate *P. aeruginosa* from the environment. Using the method and primer sets described above, we isolated 81 candidate *P. aeruginosa* cells (Appendix A). 16S rDNA sequencing indicated that all 81 of these isolates were *P. aeruginosa* cells. Thus, we were able to establish a method to isolate almost exclusively *P. aeruginosa* without sequencing. In the above series of studies, we succeeded in isolating a total of 110 strains, from which 50 strains (EN-001 to EN-050) were randomly selected for subsequent antibiotic susceptibility and risk assessment studies.

### 3.3. The Ratio of Antibiotic-Resistant Strains of P. aeruginosa in the Environment and Clinical Setting

First, in order to investigate the characteristics of *P. aeruginosa* isolated from the environmental and clinical setting, the minimum inhibitory concentration (MIC) of various antibiotics was evaluated. Since it is necessary to examine the susceptibility of environmental isolates as well as clinical isolates, tetracycline, which is commonly used in livestock production, and rifampicin, which can also be used to evaluate hypermutants, were added to the evaluation, and five representatives from each category were selected based on the WHO’s classification of antibiotics. According to Clinical and Laboratory Standards Institute Guideline M100-S22 [32], the minimum inhibitory concentration criteria for classifying resistant strains have been defined as ciprofloxacin ≥4 µg/mL, imipenem ≥8 µg/mL, aztreonam ≥32 µg/mL, and amikacin ≥64 µg/mL. For other antibiotics not specified therein, resistant strains were defined as those with MICs 8 times higher than those of PAO1S. Results are shown in Appendix A and Figure 3 environmental strains (EN-002, EN-014) showed resistance to some antibiotics, but most strains were susceptible to antibiotics. On the other hand, 15 of the 42 clinical isolates from the same area showed resistance to some antibiotics, and ultimately, the percentage of antibiotic-resistant strains was 4% in environmental isolates and 35.7% in clinical sites, indicating that the percentage of resistant strains was clearly higher in clinical sites. The characteristic feature of the resistant strains in the environmental isolates was that only one strain showed resistance to more than two antibiotics, whereas the characteristic feature of the resistant strains in the clinical isolates was that more than 66% (10 out of 15 strains) showed higher MICs against two or more antibiotics (Figure 3, Appendix A). Both the environmental and clinical isolates in this region exhibited a characteristic with a high ratio of strains resistant to chloramphenicol, and no strains showing resistance to tetracycline or rifampicin. On the other hand, with respect to imipenem, in the case of environmental strains, all strains represented sensitivity, but in the case of clinical strains, 9 out of 15 resistant strains showed resistance to imipenem, indicating a clear difference depending on the isolated location.

### 3.4. Antibiotic-Tolerance Level of P. aeruginosa in the Environment and Clinical Setting

Recent studies have revealed that, unlike antibiotic-resistance, antibiotic-tolerance is a transient phenotype exhibited only by a subset of bacteria in a clonal population, although it is an innate function of bacteria [16]. This antibiotic-tolerance has been noted to lead to chronic, refractory, and persistent infections and should be watched very closely in clinical practice. However, there is no standardized method to easily assess antibiotic-tolerance. From the viewpoint of infection control, there is an urgent need to analyze the mechanism of tolerance development of nosocomial infection-causing bacteria in various regions and to investigate their antibiotic-tolerance rates.

The following matters were considered in establishing conditions for evaluating the tolerance of *P. aeruginosa*. It has been reported that have previously reported that adherent bacteria on solid surfaces are already tolerant to antibiotics before forming biofilm [33], and that adherent and biofilm bacteria were less susceptible to several antibiotics compared to planktonic bacteria in clinical isolates from cystic fibrosis patients [34]. In addition, Murakami et al. [35] reported that presented the evidence that the accumulation of intracellular c-di-GMP (bis-(3′,5′)-cyclic-di-guanidine monophosphate) levels by surface adherence lead the antibiotic-tolerance in PAO1S. Therefore, we decided that it was appropriate to conduct tolerance evaluation using adhesion bacteria rather than planktonic bacteria.

Murakami et al. [24] showed that strains that have recently become nosocomial despite not being resistant to antibiotics have a high proportion of strains that show tolerance to antibiotics. One question arose for us: is this ratio the same for environmental commensal bacteria, which may be less exposed to antibiotics than in the clinical setting? To answer this question, we decided to investigate the antibiotic-tolerance of environmental isolates in detail following the antibiotic susceptibility evaluation.

For antibiotic-tolerance evaluation, we followed the method of Murakami et al. [24]. A total of 33 clinical strains were evaluated, excluding 9 strains resistant to imipenem from the susceptibility evaluation, and all environmental isolates were evaluated because all 50 strains were sensitive to imipenem. Antibiotic-tolerance was measured by the MBC^AD^/MIC^AD^ ratio. In this study, the ratio of 32 or higher is defined as high tolerance that same as Murakami et al. [24]. As a result, surprisingly, 45 out of 50 imipenem-sensitive environmental strains (90%) showed high tolerance (Figure 4 and Appendix A). Comparison between the populations was performed by Pearson’s chi-square test (*p* value of <0.01). The proportion of high tolerance was represented that significantly higher for environment than for clinics (*p* = 0.0012). This result can be interpreted that many environmental strains have a high tolerance, which is a strain having a risk of making treatment difficult in case of infection. In the case of clinical strains, it was found that 26 strains (78.8%) out of 33 strains showing sensitivity to imipenem represented high tolerance (Figure 4 and Appendix A). Those results mean that in the case of *P. aeruginosa* isolated in this region, it was found that a significant number of strains that has high tolerance regardless of the isolated location.

## 4. Discussion

In this study, to evaluate the susceptibility and infection risk of bacteria present in the environment, we isolated and evaluated *P. aeruginosa* cells from the environment, taking those cells, an environmental endemic bacterium, as a model bacterium from river water in the city. As the first step, we established a new method to isolate only *P. aeruginosa* from the environment that combined selects the appropriate culture temperature for the growth of *P. aeruginosa* and the PCR method using a newly designed primer based on the 16S rDNA-specific sequence of *P. aeruginosa* (Figure 2a). This method does not require special equipment, advanced expertise, and long analysis time, so it can efficiently isolate only *P. aeruginosa* from environmental samples without location or time constraints. Since antibiotic selection pressure is not applied, the degree of antibiotic-resistance of *P. aeruginosa* in the environment can be accurately measured. In order to conduct epidemiological studies on a total of 92 *P. aeruginosa* strains isolated, the POT method used as described in Section 2.6. The results showed that 92 strains that isolated in this study has a variety genotype (Appendix A). The POT method can compare the genetic homogeneity between strains through the results of Multiplex PCR using only a general PCR device, so it is able to get the results even in a limited analysis environment [27]. Furthermore, since the results are evaluated to be nearly same discriminability as the pulsed field gel electrophoresis (PFGE) and Multilocus Sequence Typing (MLST), it seems the most appropriate way to confirm the genetic characteristics of the strains isolated by our established separation method.

Subsequently, when we compared the antibiotic-resistance characteristics of *P. aeruginosa* isolated from the environment with those of *P. aeruginosa* isolated from clinical settings at the same time in the same region, the proportion of antibiotic-resistant strains was higher in clinical settings, as expected, and the proportion of strains that showed resistance to various antibiotics was also higher in clinical settings (Figure 3). In particular, the characteristics of resistance to antibiotics commonly used to treat *P. aeruginosa* in clinical settings led us to speculate that the resistance of clinical isolates likely developed due to antibiotic selection pressure. On the other hand, regarding the 50 strains isolated from the environment and analyzed, the number of antibiotic-resistant strains was significantly low, despite the different collection sites and collection times, suggesting that the rivers in this area are not very contaminated with antibiotics and antibiotic-resistant bacteria. Therefore, judging solely from the results of the MIC measurements, it can be inferred that the risk to human health from antibiotics and antibiotic-resistant bacteria in the environment in this area is very low.

Based on the MIC results, the risk of resistant bacteria in this analysis area was considered to be low. However, just to be sure, the presence of hyper-mutators with mutations in *mutS*, which is a risk for the emergence of antibiotic-resistant bacteria, was also examined according to the method of Oliver et al. [36]. As shown in Appendix A, the mutation frequency more than 100 times higher than that of PAO1S was extremely low, which could be interpreted as having a sufficiently low risk of developing resistant bacteria in this analysis area.

In recent years, resistance, as well as tolerance to antibiotics, has received increasing attention as a risk for nosocomial infections. In particular, carbapenems are one of the main antibiotics used to treat *P. aeruginosa* infection in clinical settings [37], and strains that exhibit high resistance or tolerance to carbapenems survive and avoid treatment, making treatment difficult. In fact, in the previous experiment, among *P. aeruginosa* isolated from clinical settings in this region, patients infected with carbapenem high tolerance strains were found to be highly related to severe diseases [24]. Since *P. aeruginosa* existing in the environment can infect humans at any time through various routes [38,39], environment-clinical linkage analysis is required. In addition to the need to determine the abundance of resistance, we also had a fundamental question as to whether the expression of antibiotic-tolerance is affected by prior exposure to antibiotics. Therefore, we investigated in detail the carbapenem-tolerance of strains isolated from the environment. Contrary to expectations, the percentage of highly tolerant strains was high for both environmental and clinical settings, rather slightly higher for environmental strains at 90% than for clinical isolates at 78.8%. This is the first report showing the percentage of high tolerance strains of *P. aeruginosa* of environmental origin. The results represented that *P. aeruginosa* should be considered the most at-risk bacterium to watch out for in nosocomial infections because of its very high tolerance activity.

Antibiotic-tolerance is expressed regardless of the resistance mechanism and is clearly distinguished from resistance [15,40]. Therefore, it is difficult to evaluate the antibiotic-tolerance of strains that have already developed antibiotic-resistance. Antibiotic-tolerance evaluation is considered to be one of the appropriate methods to evaluate the potential risk of bacteria that do not show antibiotic-resistance through the MIC assay, and it is expected that the risk associated with antibiotic-tolerance can be accurately identified through an appropriate combination of the two evaluation methods.

## 5. Conclusions

Since both the proportion of antibiotic-resistance bacteria and the risk of emergence of antibiotic-resistance bacteria are extremely low in the environment of this region, the risk of infection by antibiotic-resistant bacteria is considered low. However, there are many high tolerance strains in the environment, as shown in this study. It needs to introduce a system that can monitor and control bacteria in the environment and methods to reduce the risk of infection with infectious bacteria such as *P. aeruginosa*, which have a high potential antibiotic-tolerance ability. The isolation and risk assessment methods used in this study can be applied to various samples and regions and thus can be applied to country-specific risk assessment based on the analysis results in each country.

## Figures and Tables

**Figure 1 antibiotics-11-01120-f001:**
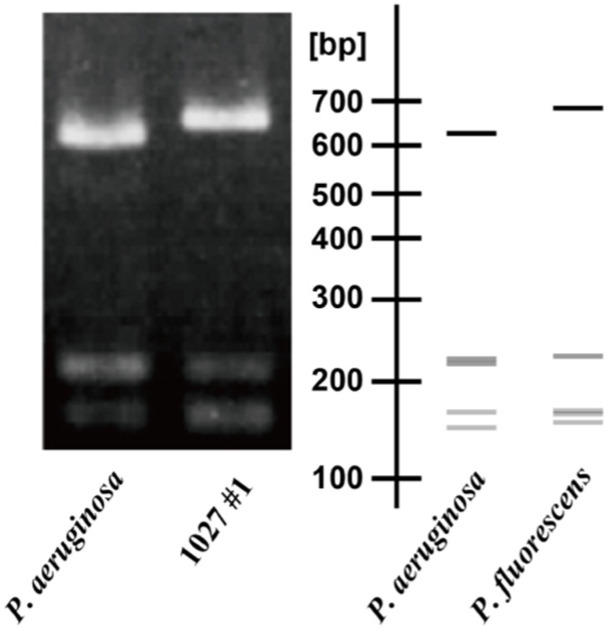
Patterns of restriction fragment length polymorphism analysis of *P. aeruginosa* PAO1S and 1027 #1 strain. Image drawing of *P. aeruginosa* and *P. fluorescens* based on NCBI’s database (**right**) and actual electrophoresis (**left**). 1027 #1 is one of 9 strains isolated by incubation using a NAC agar plate at 37 °C.

**Figure 2 antibiotics-11-01120-f002:**
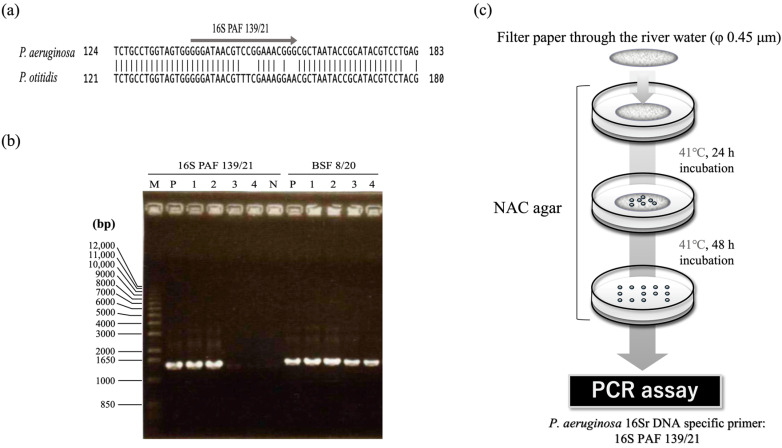
Schematic of the isolation method for *P. aeruginosa* from the environment. (**a**) Surrounding sequence of the 16S rDNA to which the primer, 16S PAF 139/21, binds, and sequence of 16S rDNA sequences of *P. aeruginosa* and *P. otitidis*. (**b**) Electrophoresis diagram of DNA fragment amplified with primer set, 16S PAF 139/21(or BSF 8/20) and BSR 1541/20 against genome DNA of *P. aeruginosa* and *P. otitidis*. Lane M: 1 kb ladder; lane P: *P. aeruginosa* PAO1S; lanes 1 and 2: *P. aeruginosa* isolated from the river in this study; lanes 3 and 4: *P. otitidis* isolated from the river in this study. (**c**) The flow of the process and cultivation conditions using NAC agar.

**Figure 3 antibiotics-11-01120-f003:**
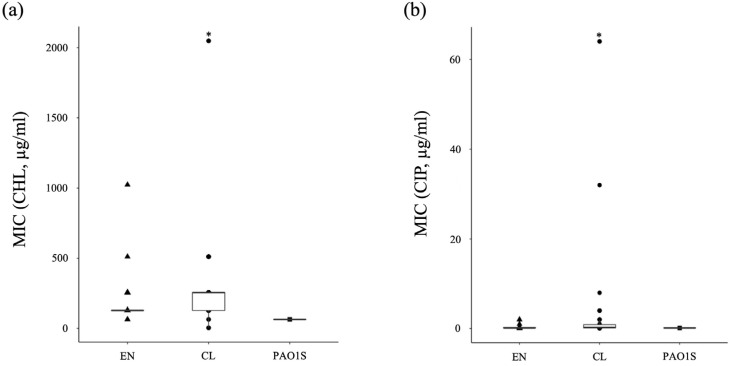
Graphical presentation of the results of antibiotic susceptibility of the strains in Appendix A. ▲: the strains isolated from the environment, EN-001-050, Appendix A. ●: the strains isolated from clinical settings, CL01-42, Appendix A. ■: PAO1S, which is the standard strain of *P. aeruginosa*. *: the MIC is equal to or higher than that concentration. (**a**) CHL, chloramphenicol; (**b**) CIP, ciprofloxacin; (**c**) TET, tetracycline; (**d**) IPM, imipenem; (**e**) ATM, aztreonam; (**f**) AMK, amikacin; (**g**) RIF, rifampicin; (**h**) MIN, minocycline.

**Figure 4 antibiotics-11-01120-f004:**
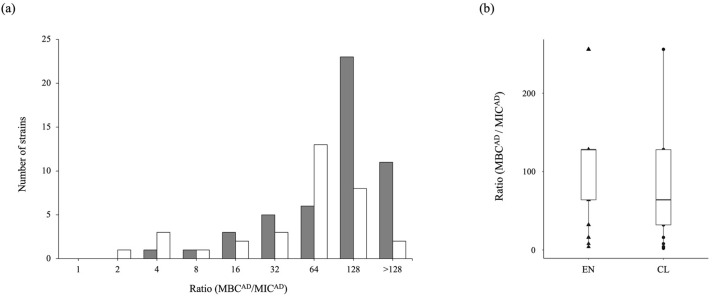
The ratio distribution of adherent bacteria MBC^AD^/MIC^AD^. (**a**) The number of strains each ratio (MBC^AD^/MIC^AD^). The gray bar represents the number of strains isolated from environment, EN-001-050, and the white bar represent the number of strains isolated from clinical setting, CL01-42. In the total number of 42 clinical strains, nine strains that exhibit resistance to imipenem were excluded. (**b**) The results were represented as a box plot to compare the MBC^AD^/MIC^AD^ ratio of environmental and clinical strains. ▲: the strains isolated from the environment, EN-001-050, Appendix A. ●: the strains isolated from clinical settings that nine strains that exhibit resistance to imipenem were excluded, CL01-42, Appendix A.

## Data Availability

The data that support the findings of this study are available within the article.

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
