# Peer review of "Characteristics of Antibiotic Resistance and Tolerance of Environmentally Endemic Pseudomonas aeruginosa"

_antibiotics, 2022, doi:10.3390/antibiotics11081120_

Round 1

Reviewer 1 Report

Reviewer

The authors submitted the article titled "Characteristics of Antibiotic Resistance and Tolerance of Environmentally Endemic Pseudomonas aeruginosa". The study presents a simple and innovative culture method to isolate P. aerugunisa strains in the environment. Next, they assessed the antimicrobial sensitivity of the isolated bacteria, determining antibiotic resistant and tolerant strains. The study is interesting. However, it needs major revisions to be published.

Introduction.

Major comments

-       The authors should mention updated data from the World Health Organization on the percentages of patient deaths annually from infections with multidrug-resistant bacteria.

-       Before introducing antibiotic tolerance, the authors should better describe the antibiotic resistance, defining the main factors that lead to the acquisition of antibiotic resistance: i) Transduction, ii) conjugation, iii) transformation and iv) horizontal gene transfer mediated by OMVs (https://doi.org/10.3390/ijms22168732).

-       Since this ability is a mechanism originally possessed by bacteria, it is thought that there must be a similar number of bacteria that exhibit antibiotic tolerance more easily in the environment, but there are no actual examples of such bacteria. This sentence is unclear. Authors should rephrase.

Materials and methods

Major comments

-       To facilitate the reproducibility of the experiments for readers, the materials used (e.g. Luria-Bertani, Agar etc.) and the instrumentation used (e.g. thermal cycler) brands should be specified. Review the entire Materials and Methods section and add the missing information

-       Section 2.3: How were clinical strains of P. aeruginosa isolated and identified? Specify in this section isolation and identification.

-       Section 2.4: minimum inhibitory concentration. The authors did not specify controls (e.g. positive control, untreated bacteria). Also, specify the concentration range used for each antibiotic.

-       Section 2.5: Why do you use adherent and non-suspended bacteria for carbapenems? Clarify the biological significance of this essay. Also, what concentration range of imipenem do the authors use? Finally define the controls in this case.

Results

Major comments

-       Section 3.1: I believe that the authors should have supported the PCR data and sequencing with another technique to be sure they had isolated P. aeruginosa only using the NAC-agar and growth at 41 ° C. It might be useful to associate an identification method based on MALDI-TOF mass spectrometry. After this further confirmation, it is possible to say with absolute certainty that this method only allows for the growth of P. aeruginosa

-       Section 3.3: why did you choose the mic 8 times as the cut-off from the reference strain? support this decision with bibliographic references.

-       Reason for the choice of antibiotics used to assess the susceptibility of P. aeruginosa strains. In my opinion, the authors should have used one antibiotic for each class, according to the EUCAST / CLSI guidelines.

-       Figure 3. I would recommend a clearer representative method, such as histograms.

-       Discussions

Minor comments:

-       Authors should comment on the results obtained with literature data. Has this method ever been used? Are there any other similar studies?

-       The low percentage of resistant strains among environmental isolates was also considered to indicate that the rivers in this area are very little contaminated by antibiotics and antibiotic-resistant bacteria: To make this consideration, the experiment had to be repeated at different moments in time.

Author Response

 Thank you for reading our manuscript and giving us your precious opinion in great detail. We agree with your points out in many parts. We have revised it so that the plaintiff can deliver more appropriate information. The details of the modifications are as follows.

 Please note that the revised manuscript does not have a line number, so the revised parts were marked using the section. Also, the revised supplementary data is added at the end of the manuscript. We appreciate checking the last part of the manuscript for revised supplementary data.

Major comments

-   The authors should mention updated data from the World Health Organization on the percentages of patient deaths annually from infections with multidrug-resistant bacteria.

 We agree with you that we should provide more specific information. The first sentence of introduction has been modified as follows.

 Antibiotic-resistant bacteria are a serious public health problem. In particular, it is reported that the number of deaths caused by multidrug-resistant bacteria worldwide reaches 700,000 a year, and unless special measures are taken, the number of deaths per year could increase to 10 million by 2050. Antibiotic-resistant bacteria are a severe public health problem [1,2].

-   Before introducing antibiotic tolerance, the authors should better describe the antibiotic resistance, defining the main factors that lead to the acquisition of antibiotic resistance: i) Transduction, ii) conjugation, iii) transformation and iv) horizontal gene transfer mediated by OMVs (https://doi.org/10.3390/ijms22168732).  

 We thank the reviewer for pointing out the lack of explanation. Also, we are glad to point out the important references that we missed. we have revised it as below.

 It has reported that the main factors that lead to the acquisition of antibiotic-resistance from the outside of bacteria cells are as follows: i) transduction [3], ii) conjugation [4,5], iii) transformation [6] and iv) horizontal gene transfer mediated by OMVs [7]. In addition, mutations that reduce the binding of antibiotics to their targets [8-10] and increased expression of efflux pumps [11-14] that export antibiotics out of cells have been identified as mechanisms of antibiotic-resistance due to change inside of bacteria cells.

-   Since this ability is a mechanism originally possessed by bacteria, it is thought that there must be a similar number of bacteria that exhibit antibiotic tolerance more easily in the environment, but there are no actual examples of such bacteria. This sentence is unclear. Authors should rephrase.

 We agree that your opinion. It was hard to understand because the content of the sentence was implicit. We rephrase the sentence as follows.

 Recently, Murakami's group analyzed Pseudomonas aeruginosa isolated from clinical settings and reported that about 60% of the strains were high-tolerance [24].

 Since antibiotic-tolerance is an intrinsic property of bacteria, it is thought that high-tolerance strains exist in similar proportions regardless of location, but it is possible that exposure to antibiotics during treatment is involved in the expression of this function.

Materials and methods

Major comments

-   To facilitate the reproducibility of the experiments for readers, the materials used (e.g. Luria-Bertani, Agar etc.) and the instrumentation used (e.g. thermal cycler) brands should be specified. Review the entire Materials and Methods section and add the missing information.

 Thank you for pointing out the missing information. We reviewed the entire materials and method section and added the information on reagents and the instruments.

-   Section 2.3: How were clinical strains of P. aeruginosa isolated and identified? Specify in this section isolation and identification.

 We acknowledge that there was a lack of detail information on how to isolate P. aeruginosa at the clinical site. The following information was added to the manuscript at Section 2.3, and additional information on the source of separation of each strain was also attached as supplementary data (Table S4).

 P. aeruginosa clinical strains were isolated in Tokushima University Hospital and this study was performed in accordance with the Declaration of Helsinki and the study protocol was approved by the Institutional Review Board of Tokushima University Hospital (approval number: 1300-1). Strain identification was carried out by an automatic biochemical identification system (MicroScan WalkAway, Beckman Coulter, Inc., California, USA). P. aeruginosa clinical strains were isolated from April to November in 2011. The sources of them were listed in the Table S4.

-   Section 2.4: minimum inhibitory concentration. The authors did not specify controls (e.g. positive control, untreated bacteria). Also, specify the concentration range used for each antibiotic.

 We apologize for the insufficient information in this section. We added the concentration range of antibiotics and information about control conditions throughout Section 2.4.

-   Section 2.5: Why do you use adherent and non-suspended bacteria for carbapenems? Clarify the biological significance of this essay. Also, what concentration range of imipenem do the authors use? Finally define the controls in this case.

 Thank you for pointing it out. We agree that further information is needed on this part. We added the following additional information and references to the appropriate section (Section 2.5 and Section 3.4) of the manuscript.

[Materials and Methods]

The final imipenem concentration was used from 0.125 mg/ml to 256 mg/ml.

[Results]

This antibiotic-tolerance has been noted to lead to chronic, refractory, and persistent infections and should be watched very closely in clinical practice. However, there is no standardized method to easily assess antibiotic-tolerance. From the viewpoint of infection control, there is an urgent need to analyze the mechanism of tolerance development of nosocomial infection-causing bacteria in various regions and to investigate their antibiotic-tolerance rates.

The following matters were considered in establishing conditions for evaluating the tolerance of P. aeruginosa. It has been reported that have previously reported that adherent bacteria on solid surfaces are already tolerant to antibiotics before forming biofilm [33], and that adherent and biofilm bacteria were less susceptible to several antibiotics compared to planktonic bacteria in clinical isolates from cystic fibrosis patients [34]. In addition, Murakami et al. [35] reported that presented the evidence that the accumulation of intracellular c-di-GMP (bis-(3’,5’)-cyclic-di-guanidine monophosphate) levels by surface adherence lead the antibiotic-tolerance in PAO1S. Therefore we decided that it was appropriate to conduct tolerance evaluation using adhesion bacteria rather than planktonic bacteria.

In this study, the ratio of 32 or higher is defined as high-tolerance that same as Murakami et al. [24].

Results

Major comments

-   Section 3.1: I believe that the authors should have supported the PCR data and sequencing with another technique to be sure they had isolated P. aeruginosa only using the NAC-agar and growth at 41 ° C. It might be useful to associate an identification method based on MALDI-TOF mass spectrometry. After this further confirmation, it is possible to say with absolute certainty that this method only allows for the growth of P. aeruginosa.

 We agree that using MALDI-TOF mass spectrometry to identify strains is useful. However, we wanted to emphasize that this study found a way to accurately isolate with P. aeruginosa specific 16s rDNA sequence, without expensive specialized equipment, using only a general PCR machine. We believe that this method will enable a uniform evaluation of resistant strains even in developing countries, where the use of expensive equipment is limited, and resistant strains are most prevalent. We have also confirmed that the POT method*, which is a multiplex PCR-based analysis technique, is sufficiently diverse, even among P. aeruginosa isolates, and have confirmed the validity of this isolation method. This method, in combination with POT, makes it possible to determine the variety of P. aeruginosa at a low cost, even in areas with limited equipment. We have added the results of the POT method to the discussion section because we believe it is necessary to clearly state one of the effectiveness of this isolation method, "isolation of Pseudomonas aeruginosa with variation”.

 POT method*: The POT method is useful because it meets the aim of this study, and it is possible to analyze it with only a general PCR machine. Also, we can get the appropriate genetic information that can compare the homogeny between strains by POT types, and it is known that they exhibit almost the same discrimination as the pulsed field gel electrophoresis (PFGE) and Multilocus Sequence Typing (MLST).

-   Section 3.3: why did you choose the mic 8 times as the cut-off from the reference strain? support this decision with bibliographic references.

 Thank you for pointing out important references that we missed. The reference and the concentration of the determination criteria of resistant strains for each antibiotic were specifically written in Section 3.3.

 According to Clinical and Laboratory Standards Institute Guideline M100-S22 [32], the minimum inhibitory concentration criteria for classifying resistant strains have been defined as ciprofloxacin ≥ 4 µg/ml, imipenem ≥ 8 µg/ml, aztreonam ≥ 32 µg/ml, and amikacin ≥ 64 µg/ml. For other antibiotics not specified therein, resistant strains were defined as those with MICs 8 times higher than those of PAO1S.

-   Reason for the choice of antibiotics used to assess the susceptibility of P. aeruginosa strains. In my opinion, the authors should have used one antibiotic for each class, according to the EUCAST / CLSI guidelines.

 Thank you for your suggestion. The reason for choosing antibiotics has been added to Section 3.3 as follows:

 Since it is necessary to examine the susceptibility of environmental isolates as well as clinical isolates, tetracycline, which is commonly used in livestock production, and rifampicin, which can also be used to evaluate hypermutants, were added to the evaluation, and five representatives from each category were selected based on the WHO's classification of antibiotics.

Discussions

    Minor comments:

-   Authors should comment on the results obtained with literature data. Has this method ever been used? Are there any other similar studies?

 There has been no report of the isolation of P. aeruginosa from the environment using the combination of PCR, culture medium, and incubation temperature that we have developed here. In addition, no detailed measurements of antibiotic tolerance of environmental commensal bacteria have been reported, and this paper is the first report of such a study.

 Therefore, in order to clearly indicate the result, the following statement was made in the text: “This is the first report showing the percentage of high-tolerance strains of P. aeruginosa of environmental origin.” in the last part of the 4th paragraph of the discussion.

-   The low percentage of resistant strains among environmental isolates was also considered to indicate that the rivers in this area are very little contaminated by antibiotics and antibiotic-resistant bacteria: To make this consideration, the experiment had to be repeated at different moments in time.

 We were sorry for not providing enough information to make it easier to understand. The 50 environmental isolates used in this paper were taken from three different spots of rivers in the same city, and there was a time difference of about three months in the resampling process after the establishment of the experimental method. Nevertheless, there was no difference between the collection place or time so that expression was used. Therefore, we have added the following sentence recognizing this limitation.

 On the other hand, regarding the 50 strains isolated from the environment and analyzed, the number of antibiotic-resistant strains was significantly low, despite the different collection sites and collection times, suggesting that the rivers in this area are not very contaminated with antibiotics and antibiotic-resistant bacteria.

Reviewer 2 Report

I appreciate the authors provided such an interesting study to reveal the environmental facts about P. aeruginosa. Only a few concerns from the clinical aspect which is my expertise:

1.      The co-authors Dr. Murakami and Dr. Azuma kindly provided the clinical isolates from the hospital. However, this procedure should come with an IRB certificate to ensure no breach of private information from patients occurred. Please provide the IRB No.

2.      Method 2.2. Please provide the collecting time for the river water samples.

3.      Method 2.3. Please provide the isolation time and source for the 42 isolates.

4.      Figure 3. It’s hard to read via a separate pattern for comparing 2 sets of data. Please consider the “dot plot for grouped data”, e.g. one antibiotic one plot. Reference as followed: https://r-graphics.org/recipe-distribution-dot-plot-multi, but not limited to construct it with R. By doing so, the statistical analysis, ANOVA, would be able to present on the plot, which would make the data more reliable.

5.      Figure 4. Also, suggest presenting in the form mentioned in comment 4.

6.      The pulse-field gel electrophoresis (PFGE) is suggested to carry out to better understand the phylogenetic relationships between these 92 isolates (50+42). 

Author Response

 We appreciate for reading our manuscript and giving us your precious comments. We agree with your comments and opinions in many parts. We have revised it so that the plaintiff can deliver more appropriate information. The details of the modifications are as follows.

 Please note that the revised manuscript does not have a line number, so the revised parts were marked using the section. Also, the revised supplementary data is added at the end of the manuscript. We appreciate checking the last part of the manuscript for revised supplementary data.

1. The co-authors Dr. Murakami and Dr. Azuma kindly provided the clinical isolates from the hospital. However, this procedure should come with an IRB certificate to ensure no breach of private information from patients occurred. Please provide the IRB No.

        Thanks to your point, we have recognized that there is a lack of information on isolation methods of clinical strains. We added the information about IRB No. to the text of the manuscript in the Section 2.3 as follows. In addition, information on the source of the clinical strain was added as supplementary data (Table S4).

P. aeruginosa clinical strains were isolated in Tokushima University Hospital and this study was performed in accordance with the Declaration of Helsinki and the study protocol was approved by the Institutional Review Board of Tokushima University Hospital (approval number: 1300-1). Strain identification was carried out by an automatic biochemical identification system (MicroScan WalkAway, Beckman Coulter, Inc., California, USA).

2. Method 2.2. Please provide the collecting time for the river water samples.

        We thank you for pointing it out. We added information about the time period when the sample was collected. We have changed the first sentence to the Section 2.2 follow as:

The river water was collected from three major riversides in downtown Tokushima, Tokushima Prefecture through June to September in 2012.

3. Method 2.3. Please provide the isolation time and source for the 42 isolates.

        We apologize for insufficient information on clinical isolated P. aeruginosa. We have updated the information in Section 2.3.

P. aeruginosa clinical strains were isolated from April to November in 2011. The sources of them were listed in the Table S4.

4. Figure 3. It’s hard to read via a separate pattern for comparing 2 sets of data. Please consider the “dot plot for grouped data”, e.g. one antibiotic one plot. Reference as followed: https://r-graphics.org/recipe-distribution-dot-plot-multi, but not limited to construct it with R. By doing so, the statistical analysis, ANOVA, would be able to present on the plot, which would make the data more reliable.

        We agree with your point. Thank you for giving us a very useful example. We modified Figure 3 to make it easier to see that was modified to be expressed as a box plot using R. The modified figure shows the characteristics of the environment strains and clinical strains for each antibiotic. The figure has been simplified by using the box plot, but it makes easy to understand because it represented various statistical information such as quartile, interquartile range, median, and outliers.

5. Figure 4. Also, suggest presenting in the form mentioned in comment 4.

        As you pointed out in Figure 3, we made additional Figure 4 (b) to make it easier to compare the results between environmental strains and clinical strains. Also, we did Pearson’s chi-square test. Statistical analysis was performed by JMP software 13 (SAS Institute, Inc., Tokyo, Japan). The proportion of high-tolerance was significantly higher for environment than for clinics (p=0.0012). P value of < 0.01 was considered as statistically significant.

6. The pulse-field gel electrophoresis (PFGE) is suggested to carry out to better understand the phylogenetic relationships between these 92 isolates (50+42). 

        We appreciate the reviewer's insightful suggestion and agree that it would be useful to add the analyze the phylogenetic relationship between these 92 isolates. Therefore, we conducted an analysis using POT (PCR-based OFR Typing), a multiplex PCR analysis technique, to do a molecular epidemiological analysis on 92 isolates. Since the POT method is a useful because it meets aim of this study and it is possible to analyze it with only a general PCR machine. Also, we can get the appropriate genetic information that can compared the homogeny between strains by POT types and it is known that exhibit almost the same discrimination as the pulsed field gel electrophoresis (PFGE) and Multilocus sequence typing (MLST).

 In this study, the main goal is to isolate the P. aeruginosa using only general PCR machines without expensive specialized equipment and to identify the properties of their antibiotic tolerance in the environment. In particular, we believe that this method will enable a uniform evaluation of resistant strains even in developing countries, where the use of expensive equipment is limited, and resistant strains are most prevalent. We have confirmed that using the POT method, a multiplex PCR-based analysis method, we verify that there is sufficient variation between 92 strains isolated in our study. This method, in combination with POT, makes it possible to determine the variation of P. aeruginosa at low cost, even in areas with limited equipment. We have added the results of the POT method to discussion section because we believe it is necessary to clearly state one of the effectiveness of this isolation method, "isolation of Pseudomonas aeruginosa with variation”.

Reviewer 3 Report

Following are my comments.

1. Abstract is very simple and the results section is the more likely conclusion, authors must provide the main results in the abstract result section. 

2. Please check the introduction for grammatical and spelling errors. The objective of the study must be elaborated on at the end of the introduction part. How about the reported data on the Antibiotic Resistance and Tolerance of Environmentally Endemic Pseudomonas aeruginosa from LMICs and developed countries, please provide some data from other parts of the world. ''We have been trying..... , We isolated P. aeruginosa..... please avoid the word 'we' in the introduction part and can be exchanged with the appropriate word.

3. How about the analysis of the data, Are any statistical or descriptive results drawn? How the data were taken in the form of graphs instead tables or over-time analysis.

4.  Authors must compare the present results with reported data as the discussion is lacking such data. 

5. Give a separate heading for the conclusion section.

Good Luck

Author Response

 We appreciate for reading our manuscript and giving us your precious comments. We agree with your points out in many parts. We have revised it so that the plaintiff can deliver more appropriate information. The details of the modifications are as follows.

 Please note that the revised manuscript does not have a line number, so the revised parts were marked using the section. Also, the revised supplementary data is added at the end of the manuscript. We appreciate checking the last part of the manuscript for revised supplementary data.

1. Abstract is very simple and the results section is the more likely conclusion, authors must provide the main results in the abstract result section.

 We revised the sentence that added the following information about the results in the abstract.

 The 50 strains were randomly selected among 110 isolated from the river. The results of antibiotic susceptibility evaluation showed that only 4% of environmental strains were classified as antibiotic-resistant, while 35.7% of clinical strains isolated in the same area were antibiotic-resistant, indicating a clear difference between environmental and clinical strains. However, the percentage of antibiotic-tolerance, an indicator of potential resistance risk for strains that have not become resistant, was 78.8% for clinical strains and 90% for environmental strains, suggesting that P. aeruginosa, known cause of nosocomial infections, has high rate of antibiotic-tolerance even in environmentally derived strains.

2. Please check the introduction for grammatical and spelling errors. The objective of the study must be elaborated on at the end of the introduction part. How about the reported data on the Antibiotic Resistance and Tolerance of Environmentally Endemic Pseudomonas aeruginosa from LMICs and developed countries, please provide some data from other parts of the world. ''We have been trying..... , We isolated P. aeruginosa..... please avoid the word 'we' in the introduction part and can be exchanged with the appropriate word.

 Thank you for suggesting that we revise the introduction part. We added more specific information in the first part of the introduction and revised some parts of the sentences (last paragraph of the introduction) according to your and other reviewers' comments.

3. How about the analysis of the data, Are any statistical or descriptive results drawn? How the data were taken in the form of graphs instead tables or over-time analysis.

 We agree that statistical work and graphic modifications are needed to increase the reliability of our data and make the results easier to understand. Therefore, the following tasks were performed, and the manuscript was amended.

  In Fig. 3 (results of antibiotics susceptibility) was modified to be expressed as a box plot using R. The modified figure shows the characteristics of the environment strains and clinical strains for each antibiotic. The figure has been simplified by using the box plot, but it makes easy to understand because it represents various statistical information such as quartile, Interquartile range, median, and outliers.

 In Fig. 4, the MBCAD/ MICAD ratio of 32 or higher is defined as high-tolerance because that of PAO1S is 32. The proportion of high-tolerance was significantly higher for environment than for clinics (p=0.0012). Comparison between the populations was performed by Pearson’s chi-square test. Statistical analysis was performed by JMP software 13 (SAS Institute, Inc., Tokyo, Japan) and a P value of < 0.01 was considered as statistically significant.

4. Authors must compare the present results with reported data as the discussion is lacking such data. 

 There has been no report of the isolation of P. aeruginosa from the environment using the combination of PCR, culture medium, and incubation temperature that we have developed here. In addition, no detailed measurements of antibiotic tolerance of environmental commensal bacteria have been reported, and this paper is the first report of such a study.

 Therefore, in order to clearly indicate the result, the following statement was made in the text: “This is the first report showing the percentage of high-tolerance strains of P. aeruginosa of environmental origin.” in the last part of the 4th paragraph of the discussion section.

5. Give a separate heading for the conclusion section.

 Thank you for pointing it out. We deleted the last part of the discussion and created a new section, the conclusion.

Reviewer 4 Report

The authors study the antibiotic resistance and tolerance of environmentally endemic strains of Pseudomonas aeruginosa. The study is well-written but scientific interest of the results is very low and the research design is to review.

Author Response

 The authors study the antibiotic resistance and tolerance of environmentally endemic strains of Pseudomonas aeruginosa. The study is well-written but scientific interest of the results is very low and the research design is to review.

 Thank you for your suggestion. In recent years, there has been an increase in the number of reports on “Tolerance”. To support this, the definition and concept have been covered in top journals, Science in 2017 and Nature reviews microbiology in 2019. By analyzing Pseudomonas aeruginosa of environmental origin, we have shown the percentage of high-tolerance strains in P. aeruginosa that apparently have less exposure to antibiotics than clinical-derived strains, indicating that a certain number of high-tolerance strains are already present in the environment. The control of bacterial resistance is still a challenge. The control of bacterial resistance is still a very serious problem, and reports on tolerance, which is thought to be involved in such control, will become increasingly important and we would very much like to report this data in a peer-reviewed journal.

Round 2

Reviewer 1 Report

The authors answered all critical points in detail. For this reason, the manuscript can be accepted

Author Response

The authors answered all critical points in detail. For this reason, the manuscript can be accepted

Thank you very much for reviewing and accepting our manuscript. We are attaching a file that reflects the modifications made to the minor revision according to the editor's comments. We appreciate your precious comments and suggestions to improve our manuscript.

Reviewer 3 Report

The authors did not provide accurate responses, please look again to all the comments and make changes accordingly. 

Author Response

The authors did not provide accurate responses, please look again to all the comments and make changes accordingly. 

We appreciate for reviewing our manuscript again. We rechecked the manuscript and tried to respond appropriately to the reviewer's comments.

We have modified the Abstract part to represent the results, and the Introduction part was modified to show the research objectives using the appropriate citation. In addition, the result figures (Fig. 3 and 4) were modified for clarity, and the raw data were provided as supplemental data (Table S2 and S3). We have revised the conclusion section that summarizes the content to be independent. We had no chance to compare those results directly with other studies because similar studies have not been reported so far. To complement this, we modified the Discussion part by presenting additional experiment results to support the validity of our findings.

Furthermore, following the editor's comments, we made minor revisions. As a result, we believe that the manuscript has been enriched and more balanced, and we hope to publish it in the journal Antibiotics.

Reviewer 4 Report

Paper accepted

Author Response

Paper accepted

Thank you for reviewing our manuscript again. Also, we appreciate your understanding of the necessity of our manuscript and deciding to accept this manuscript. We are attaching a file that reflects the modifications made to the minor revision according to the editor's comments.